# Stressors and Impact of the COVID-19 Pandemic on Vulnerable Hispanic Caregivers and Children

**DOI:** 10.3390/ijerph20031838

**Published:** 2023-01-19

**Authors:** Mary Rodríguez-Rabassa, Estefanía Torres-Marrero, Pablo López, Kamalich Muniz-Rodriguez, Marilyn Borges-Rodríguez, Allison A. Appleton, Larissa Avilés-Santa, Luisa I. Alvarado-Domenech

**Affiliations:** 1Clinical Psychology Program, Ponce Health Sciences University, P.O. Box 7004, Ponce, PR 00732, USA; 2RCMI Center for Research Resources, Ponce Health Sciences University, P.O. Box 7004, Ponce, PR 00732, USA; 3Department of Pediatrics, Ponce Health Sciences University, P.O. Box 7004, Ponce, PR 00732, USA; 4Ponce Research Institute, Ponce Health Sciences University, P.O. Box 7004, Ponce, PR 00732, USA; 5Department of Epidemiology and Biostatistics, University at Albany School of Public Health, State University of New York, 1 University Place, Rensselaer, NY 12144, USA; 6Division of Clinical and Health Services Research, National Institute on Minority Health and Health Disparities, 6707 Democracy Blvd. Suite 800, Bethesda, MD 20892, USA

**Keywords:** COVID-19 impact, caregivers, post-traumatic stress disorder symptomatology, hair cortisol, vulnerable children, Hispanic

## Abstract

Psychological sequelae are important elements of the burden of disease among caregivers. Recognition of the impact of adversity and stress biomarkers is important to prevent mental health problems that affect rearing practices and child well-being. This cross-sectional study explored social determinants of health (SDoH)-mediated stressors during COVID-19 and risks for mental health problems among caregivers of children with prenatal Zika virus exposure. Twenty-five Hispanic caregivers completed surveys assessing SDoH vulnerabilities, COVID-exposures and impact, post-traumatic stress disorder (PTSD) symptomatology, and provided a hair sample for cortisol concentration (HCC). Most caregivers had low education, household income < $15,000/year, and were unemployed. Stressors included disrupted child education and specialized services, and food insecurity. While most reported PTSD symptomatology, multivariate linear regression models adjusted for the caregiver’s age, education, and the child’s sex, revealed that caregivers with high symptomatology had significantly lower HCC than those with low symptomatology and those with food insecurity had significantly higher HCC than participants without food insecurity. The impact of COVID-19 on daily life was characterized on average between worse and better, suggesting variability in susceptibility and coping mechanisms, with the most resilient identifying community support and spirituality resources. SDoH-mediators provide opportunities to prevent adverse mental health outcomes for caregivers and their children.

## 1. Introduction

The COVID-19 pandemic has been an unprecedented and persistent population-wide stressor with direct and indirect effects on the environment in which children live and learn, and on the quality of their relationships with caregivers, factors that contribute to their cognitive, emotional, and social development [1]. Globally, the pandemic has impacted people and societies in detrimental ways and has revealed extensive socioeconomic impacts and multilevel challenges [2]. The COVID-19 arrival to the United States (U.S.) mainland and Puerto Rico in 2020 caused severe alteration to families’ daily routines as nationwide shelter-in-place recommendations took hold, schools and childcare centers closed, and a significant portion of the workforce shifted to remote work from home [3]. Since this early period, the adult population has experienced harsh physical and psychological conditions from the infectious threat and the mitigation efforts put in place. Child caregivers have endured the burden of financial challenges from work disruption, family relational conflicts with changes in learning and daily self-care routines, and COVID-19 specific stressors, such as distressing news, limited reliable information, and difficult access to essentials [4]. In fact, evidence from this early mitigation period indicates that stressors were heightened for those caring for children, with greater challenges reported for those living in low-income and crowded households [5]. Therefore, COVID-19 has represented a traumatic event that could increase vulnerability to developing PTSD [6].

Life course research has consistently illustrated that prolonged and persistent experiences of adversity and toxic (severe and chronic) stress during sensitive life periods shape children’s brain development and health [7,8]. Biomarkers of these stressors reveal signs of physiologic dysregulation of the Hypothalamic–Pituitary–Adrenal (HPA) axis and cortisol levels with lifelong health consequences. In fact, adversity experiences have been significantly associated with hair cortisol concentration (HCC), and since cortisol accumulates in hair, this measure represents a useful marker of HPA regulation over time and of toxic stress [9].

The American Academy of Pediatrics’ policy statement on the prevention of childhood toxic stress explicitly promotes nurturing and stable home environments, and a sense of safety as buffers against children’s adversity-related health consequences [10]. In addition, evidence supports the facts that a quality adult caregiver and quality childhood relationships have a significant impact on the cognitive, emotional, and social development of children [1]. In contrast, published evidence confirms that vulnerable (i.e., at high risk of toxic stress and its consequences) caregivers during COVID-19 experienced heightened psychological distress symptoms and perceived higher levels of stress in children affecting the protective relationships [3]. In this setting, caregivers’ own coping and self-care practices model coping responses in children, threatening their susceptibility to toxic stress, as previously documented [11,12]. Therefore, it is imperative that we continue to learn more about how COVID-19 affects caregivers, and if adverse consequences can be prevented to protect children [13], especially those from populations that experience health disparities.

COVID-19, a population-wide adversity, was preceded by catastrophic natural disasters that plagued Puerto Rico, a U.S. territory with a predominately Hispanic/Latino population. In recent years, these catastrophic natural disasters included the 2017 Category 4 Hurricane Maria and the 2019 6.4 magnitude earthquake [14,15,16,17]. In September 2022, the pandemic’s impact was intensified by major infrastructural disruptions caused by the Category 1 Hurricane Fiona. According to 2022 estimates of the 2020 U.S. Census, the population of Puerto Rico is 3,221,789 with 16.7% in the 0–18 years age group [18]. From the socioeconomic perspective, census data for 2021 indicates a median household income of $21,967 and 40.5% of persons living in poverty, a percentage almost four times greater than in the U.S. Mainland (11.6%) [19]. Indeed, 58% of Puerto Rican children live below the poverty level [20]. The Study of Life Experiences, Adversities, and Resilience (SOLEAR) seeks to describe caregivers’ experiences with COVID-19 within the context of natural disasters and socioeconomic disadvantages that threaten the well-being of children and exacerbate pre-existing health or healthcare disparities. This study aimed to provide an overall descriptive and correlative analysis of distinct social determinants of health (SDoH) and measures of stress and resilience levels. The participants were caregivers and children from the Pediatric Outcomes of Prenatal Zika Exposure (POPZE) cohort study who had endured the Zika epidemic health and developmental risks and now had to confront COVID-19 daycare and parent employment disruptions that preferentially affected children with special healthcare needs, from low-income families, and from racial and ethnic minority groups [21], whose life experiences could inform critical public health decisions and actions.

## 2. Materials and Methods

Data from this cross-sectional study were collected from October through December 2020. The study involved a collection of: social determinants of health factors, participants’ experience with COVID-19-related exposures and its impacts, self-reported depression symptomatology, resilience, post-traumatic stress disorder symptomatology, and hair cortisol concentration, as described below.

### 2.1. Participants and Procedures

The study participants were caregivers enrolled from the POPZE study, which aimed to characterize the full spectrum of functional and structural abnormalities of children born to mothers with evidence of Zika virus infection during pregnancy, and to explore environmental correlates with impact on the developing brain, and health equity [22]. POPZE participants agreed to be considered for future studies. The team enrolled 25 out of 53 caregivers from POPZE, most (92%) of whom were mothers. Participants completed the study questionnaires through the REDCap [23] platform online or in person, based on their preferences. The team collected a hair sample from participants at the study site.

### 2.2. Instruments

PhenX Toolkit’s Social Determinants of Health Core (SDoH-Core) [24]. The SDoH-Core explores factors linked with social determinants of health, such as age, educational attainment, employment status, annual family income, health insurance coverage, address, food insecurity, race, and ethnicity. The team also explored spirituality (SDoH-Individual, item 30 of the WHOQOL SRPB [25]) and developed two Yes/No questions to explore social support: “I have had the support of my family to deal with this crisis” and “I have had the support of my friends and neighbors to deal with this crisis”.

COVID-19 Exposure and Family Impact Survey (CEFIS) [26]. The CEFIS assesses the effects of COVID-19 on families and caregivers via three scales: Exposure, Impact, and Distress. The Exposure Scale includes 25 “Yes/No” items exploring whether families have been exposed to pandemic-related events (e.g., decreases in family income, disruption in children’s education, and difficulties getting healthcare; α = 0.80). Participant responses endorsed as “Yes” were summed to compute a total score (range: 1–25), wherein higher scores indicated greater exposure to pandemic-related events. The 10-item Impact Scale measures the perceived impact of the pandemic on family interactions (e.g., parenting) and emotional well-being (e.g., mood) through a 4-point Likert scale (1 = Made it a lot better; 2 = Made it a little better; 3 = Made it a little worse; 4 = Made it a lot worse; α = 0.92). Response scores were averaged to compute the Impact Scale score. The Distress scale consists of two 10-point scales (1 = No distress through 10 = Extreme distress) assessing distress experienced by caregivers and children due to the pandemic (α = 0.76). The Distress scale score was computed by averaging the items’ scores.

Patient Health Questionnaire (PHQ-8) [27]. The PHQ-8 is an eight-item screening scale used to assess depression symptomatology in the general population (α = 0.82). Participants were asked to identify, in the past two weeks, how often they had been disturbed by symptoms such as “little interest or pleasure in doing things” using a four-point scale (0 = not at all; 1 = several days; 2 = more than half the days; 3 = nearly every day). Scores were summed for a total score ranging 0 to 24 points, which could be classified as not significant (0 to 4); mild (5 to 9); moderate (10 to 14); moderately severe (15 to 19); and severe (20 to 24) depression symptomatology [27]. A cutoff score of 10 or more suggested clinically significant depression symptomatology [27].

Brief Resilience Scale (BRS) [28]. The six-item BRS assesses the ability to bounce back or recover from stress (α = 0.83). Three items (1, 3, and 5) are positively worded and three (2, 4, and 6) are negatively worded. Participants were asked to indicate how much they disagreed or agreed with statements such as “it does not take me so long to recover from a stressful event” using a 5-point scale (1 = strongly disagree; 2 = disagree; 3 = neutral; 4 = agree; 5 = strongly agree). Items negatively worded were reverse coded and all items were summed to obtain the mean. Mean scores ranged from 1 to 5, with higher numbers indicating higher resilience. Scores < 2.99 were classified as low resilience, 3.00–4.30 as normal resilience, and >4.31 as high resilience [29].

Impact of Event Scale-Revised (IES-R) [30]. The IES-R is a 22-item screening instrument used to assess post-traumatic stress disorder symptomatology (α = 0.97) [31]. Participants were asked to identify how distressed or bothered they had felt during the past seven days in response to COVID-19 stressors (e.g., I felt watchful or on-guard, I tried to remove it from my memory, and I was jumpy and easily startled). Each item was rated on a five-point scale (0 = Not at all; 1 = A little bit; 2 = Moderately; 3 = Quite a bit; 4 = Extremely). A score of 33 and above indicated a probable diagnosis of PTSD [31].

Hair Cortisol Concentration (HCC). Hair has been shown to be a reliable and non-invasive method to measure chronic stress [32,33]. The team collected hair samples from the participants to assess the HPA regulatory history in response to chronic stressors via hair cortisol concentration. The cortisol assay followed the protocol developed by Meyer et al. [34], with minor modifications. Briefly, the samples were collected from the posterior vertex as close to the scalp as possible. The hair segments from the 3 cm proximal to the scalp were weighed, and approximately 35 mg were used to assess persistent stress (3 months), according to protocol. Pre-weighed hair was placed in a 5 mL polypropylene centrifuge tube to perform the wash procedure with high-performance liquid chromatography (HPLC)-grade isopropanol three times in brief intervals [35]. The wash cycle consisted of mixing hair for 3 min with 1.5 mL of the solvent in a rotator. The hair was then dried thoroughly at room temperature for at least one day. After the hair was completely dry, it was finely cut, and 1.5 mL of methanol (HPLC-grade) was added to the tube to extract cortisol during 24 h in a rotator. The methanol was transferred to 2 mL tubes and evaporation was performed using a Labconco centrivap concentrator with cold trap (Labconco, Kansas City, MO, USA). The assay buffer (250 µL) of the detection kit was used to reconstitute the cortisol, and samples were kept at −20 °C until analysis. Hair cortisol detection and quantification were possible by using the Human Skin Magnetic Bead Panel kit (MILLIPLEX MAP—Analyte Cortisol) (EMD Millipore, Chicago, IL, USA) and Luminex MAGPIX detection system (Luminex Corp., Austin, TX, USA), following the manufacturer’s instructions.

### 2.3. Statistical Analysis

Descriptive analyses were performed for sociodemographic variables, each of the instruments used (SDoH-Core, CEFIS, IES-R), and cortisol levels in the hair sample. Measures of central tendency and measures of dispersion were calculated for continuous variables. Frequencies and percentages were calculated for categorical variables. In addition, the distribution of continuous variables was analyzed, using the Skewness, Kurtosis, and Shapiro–Wilk tests. A multivariable regression analysis was conducted to describe the relationship between Cortisol Levels (HCC) and stressors. Continuous HCC levels were the outcome or dependent variable in the linear regression model. Due to the skewed distribution of the variable, a log transformation in the linear model was conducted [36]. The regression analysis models were controlled for caregivers’ age, education, and the child’s sex. Statistical analyses were performed using SPSS 28.0 (IBM Corp., Armonk, NY, USA) [37].

## 3. Results

Twenty-five caregivers completed all study questionnaires. Of these, 23 (92%) agreed to provide a hair sample for cortisol analysis.

Social Determinants of Health. Participants had a mean (standard deviation) age of 30.4 (6.6) years and identified themselves as Hispanic (100%) (Table 1). Twenty-three (92%) caregivers were female. Most of them (n = 19, 76%) considered their race as “Other” (a category that includes Puerto Rican as an option), were married or cohabitating (n = 19, 76%), and had no history of mental health problems (n = 22, 88%). Most reported having support to deal with crises from family (n = 23, 92%) and friends or neighbors (n = 20, 80%), and finding spiritual strength in difficult times (n = 20, 80%). Sociodemographic vulnerabilities were identified in a majority involving public insurance coverage (n = 20, 80%), low educational attainment (technical degree or less; n = 19, 76%), annual household income less than $15,000 (n = 19, 76%), and unemployment (n = 17, 68%). Ten caregivers (40%) reported family experiences of food insecurity. Other vulnerabilities were identified in the participants’ POPZE child, who had a mean (SD) age of 2.8 (0.7) years. Twenty-one children had a POPZE professional developmental assessment with the Bayley Scales of Infant and Toddler Development, Third Edition (BSID-II) at 24 months [22], of which 13 (61.9%) had developmental delay in at least one domain of the BSID-III: cognitive, language, or motor [38]. The authors described a decrease in prevalence of developmental delay in some domains at 36 months, indicating unaccounted protective factors, which were enacted during COVID-19. Notably, 13 of the 25 (52%) children enrolled in SOLEAR were receiving specialized services during the first pandemic year (i.e., speech, occupational, physical, or psychological therapy) of which 46.2% (n = 6) suffered service interruptions; thus, limiting remedial services needed to support their development.

COVID-19 Exposure and Family Impact Survey (CEFIS). Participants endorsed 21 out of 25 items of the Exposure scale, with a mean (SD) score of 8.84 (2.93) and a range of 4 to 14 exposures to pandemic-related events, confirming stressful experiences for all families, with some having greater exposure than others. Critical adverse exposures concerned the family’s and caregivers’ activities (Figure 1), where all participants (100%) reported stay-home orders and the closure of their child’s school or childcare. The majority referred to interruptions in their child’s education (n = 22, 88%), being unable to visit or care for a family member (n = 18, 72%), and missing an important family event (n = 16, 64%). Most of the participants reported that someone in the family was a healthcare provider or staff, or first responder (n = 18, 72%), and experienced a decrease in household income (n = 13, 52%) due to the pandemic. Of note, some participants reported a family member had been exposed to (n = 7, 28%), or contracted COVID-19, or presented symptoms (n = 6, 24%). However, none (0, 0%) experienced severe consequences (hospitalization, intensive care, or death). The mean (SD) score on the CEFIS Impact scale was 2.9 (0.48) (Figure 2), suggesting mixed implications between a little bit better and a little bit worse. While most of the pandemic’s impacts identified by caregivers were in the realms of adversity, some caregivers reported gains, such as a slightly increased perception in their ability to care for their POPZE child. Yet, results also illustrated a significant impact on the caregiver’s well-being, with reports of worsened anxiety, mood, sleeping, and exercising. Caregivers also referred to some impact on their eating habits and ability to get along with family members. Figure 3 shows that caregivers experienced heightened levels of psychological distress and perceived some distress in their children due to the pandemic.

Patient Health Questionnaire (PHQ-8). The mean (SD) of the PHQ-8 was 5.56 (4.2), a score classified as mild depression symptomatology in the previous two weeks. Most caregivers (n = 21, 85%) had scores among the mild or non-significant classification (Figure 4a); however, four (16%) had scores that suggested clinically significant depression symptomatology.

Brief Resilience Scale (BRS). Participants’ mean (SD) score in the BRS of 3.21 (0.71) reflected the expected ability to bounce back or recover from stress (normal resilience). Two-thirds of the caregivers (n = 17, 68%) had mean scores considered to indicate normal or high resilience, even in the presence of the notable stressors previously described (Figure 4b).

Impact of Event Scale-Revised (IES-R). The mean (SD) score of the IES-R was 41.08 (16.8), which was higher than the cutoff score of 33 suggested for a probable diagnosis of post-traumatic stress disorder (PTSD) [31]. Caregivers were instructed to report symptoms in response to COVID-19 stressors. Fifteen caregivers (60%) had scores at or beyond, the cutoff of 33, suggesting increased PTSD symptomatology (Figure 5) due to COVID-19 stressors. 

Hair Cortisol Concentration (HCC). The median of HCC was 75.6 pg/mg. The study team explored the associations between HCC, the CEFIS Exposure and Impact Scales, PHQ-8 depression symptoms, IES-R PTSD symptomatology, and BRS resilience scores through Pearson’s correlation analysis. As shown in Table 2, depression symptomatology was positively associated with exposure to COVID-related events (*r* = 0.53, *p* < 0.01) and PTSD symptomatology (*r* = 0.43, *p* < 0.05) scores. However, resilience was negatively correlated with depression and PTSD symptomatology scores. The team also explored the associations between HCC and stressors as measured by the IES-R and the SDoH food insecurity indicators through linear regression models, adjusted by caregivers’ age, education, and the child’s sex (Table 3). Caregivers with high IES-R scores (suggesting trauma) had significantly lower HCC (B = −1.11, SE = 0.51, *p* = 0.04). Those with food insecurity had significantly higher HCC (B = 1.01, SE = 0.48, *p* = 0.049).

## 4. Discussion

This study illustrated that COVID-19 disrupted the life of vulnerable Hispanic caregivers and children who experienced an earlier epidemic threat of serious pediatric consequences, causing important changes in their daily routines that had biological and psychosocial implications. Participants’ social determinants of health, such as low income, food insecurity, low educational attainment, and unemployment might increase their vulnerability to health disparities with a potential impact on mental health.

The most significant stress exposures due to COVID-19 were associated with the lockdown mitigation orders (i.e., school closures). A high proportion of participants reported a family member working outside the home. This phenomenon was observed in Hispanic/Latino populations in the United States mainland, who are the workers most likely to hold frontline occupations and have limited opportunities for remote work compared with other populations [39].

The lockdown involved the closure of schools and childcare facilities, a situation experienced by all study participants. Children’s education was interrupted, and caregivers took on the responsibility of assisting and supervising online education. Work disruption difficulties were identified in caregivers of children under five years of age from a nationally representative sample, where Easterly et al. found that employment disruption of caregivers increased during COVID-19 [21]. The study also found that employment disruption was significantly higher among caregivers of children with special healthcare needs, those with minority backgrounds, low-income families, and single-parent families.

In general, the caregiver’s perception of stressors was that COVID-19 had an adverse effect on their life experiences with worsened mental health and lifestyle and altered family interactions. These reports coincided with findings of COVID-19’s impact on caregivers that revealed associations between the stressors with worsened sleep disorders, stress, and social conflicts, as well as with mental health symptoms, anxiety, depression, and PTSD symptomatology [11,40,41,42,43]. The COVID-19 impact responses also suggested positive factors for families. For example, caregivers did not experience severe health consequences, such as hospitalization, intensive care, or death, during this harsh mitigation period, and some perceived improvement in their ability to care for their POPZE child.

Of note, participants’ mental health might have been impacted by school closures, since a study by Deeb et al. reported that remote learning caused caregivers to experience being overwhelmed, which was linked with increased mental health symptoms, anxiety, depression, and stress [41]. Interestingly, most participants showed none or minimal depression symptomatology, although PTSD symptomatology was highly prevalent. The prevalence of clinically significant PTSD symptomatology (60%) in this study was higher than previously reported on COVID-related studies with caregivers from different regions at different phases of the pandemic: 3.5% (China; February 2020) [44], 29% (Italy; March–May 2020) [45], 19% (United States; June–July 2020) [3], and 34% (New York; May 2020–April 2021) [41]. Although participants were instructed to report PTSD symptoms in response to COVID-19 stressors, increased symptomatology might also reflect exacerbation of symptoms experienced with previous stressors. A study with low socioeconomic status parents of school-age children found that parents’ best predictor of post-pandemic depression was having a previous history of depression [46]. Therefore, the low prevalence of depression symptomatology in caregivers in this study might be due to the absence of mental health history in most participants. It should be noted that most participants showed normal or high resilience, which may serve as a protective factor against depression and distress [28]. Indeed, in this study, resilience was negatively correlated with depression and PTSD symptomatology.

Of interest, linear regression models, adjusted by caregivers’ age, education, and the child’s sex revealed associations between PTSD symptomatology and significantly lower HCC. On the contrary, higher HCC was linked considerably with food insecurity. Cortisol is the main hormonal mediator of the body’s adaptation to perceived stress, exerting a protective life function. In the presence of toxic stress, HPA axis dysregulation of the cortisol response causes enhanced or blunted responses with abnormal levels driving organ–system pathology [47]. The ability to directly compare HCC values with other studies is limited due to variability in “normal” ranges identified in the literature and many diseases and societal factors that alter cortisol values [48,49,50]. However, these results revealed dysregulated cortisol levels in response to COVID-19 stressors, with lower HCC associated with predominantly persistent stressors.

Interestingly, other studies comparing individuals with PTSD vs. non-PTSD indicated that the disease is associated with lower HCC [51,52]. The assumption is that HCC tends to increase shortly after the traumatic event and then declines due to decreased sensitivity of the HPA axis [53]. This process could explain the HCC findings in this study, especially when considering the context in which participants have endured repeated and severe population-wide stressors, such as the Zika epidemic in 2016–2017, the Category 4 Hurricane Maria in 2017, the 6.4 earthquake in January 2020, and the COVID-19 pandemic in 2020. Therefore, the results might also reflect the biological vulnerability to chronic stressors faced by Hispanic caregivers living in Puerto Rico.

Study limitations include the cross-sectional nature of the design, which limited the ability to establish causality. The small sample size and the focus on POPZE caregivers limited the generalization of the results. The assessments were performed during a limited period, seven to nine months after the pandemic lockdown in Puerto Rico; this approach limited capturing the caregivers’ experience during the initial phase of the pandemic and the long-term impact on caregivers. However, the beginning of the pandemic represented a strenuous period with dramatic changes that could characterize other phases of the most persistent pandemic in recent years. Future studies should consider a longitudinal approach integrating stress-reduction interventions and biomarkers as objective measures of stress.

Despite study limitations, this study provides evidence of the impact of COVID-19 on a unique population of vulnerable Hispanic caregivers. Previous studies describing the exposures and impact of COVID-19 on caregivers of children include parents with children at the community level and of specific groups with chronic pain, hematology/oncology diagnoses/stem cell transplant, diabetes, and overweight/obesity issues [26,54,55] and those caring for children with developmental disabilities [56]. Data about Hispanic representation in these studies is limited. To the best of our knowledge, this is the first study describing COVID-19 exposures and impact on caregivers of children exposed to an earlier epidemic threat (Zika virus prenatal exposure). These caregivers belong to an ethnically homogeneous Hispanic group with multiple pre-existing vulnerabilities and heightened needs.

## 5. Conclusions

Hispanic caregivers of the SOLEAR study reported COVID-19 exposures consistent with families’ experiences in the United States and other countries during the harshest period of universal lockdown and subsequent periods of on–off mitigation strategies. Caregivers described an impact on daily life activities characterized, on average, as bidirectional, between worse and better, suggesting a differential susceptibility to the stressors and the presence of protective factors of endurance and recovery in some. In addition to the SDoHs identified as risk factors for poor health outcomes [57], caregivers presented other vulnerabilities, due to population-level adverse experiences in Puerto Rico and their children’s specialized healthcare needs. Children with special needs experienced disruption in remedial services. This had severe implications for children’s development and could add stress to the caregivers, who, along with their children, were exposed to the Zika virus infection and were experiencing the impact of a second health-threatening event.

COVID-19 added stressors to caregivers with psychological and physiological implications. Findings on PTSD symptomatology were above the expected levels in this vulnerable population and may have been magnified due to cumulative stressors they have faced in recent years. The presence of significant post-traumatic stress symptomatology, and its association with low cortisol levels, was consistent with findings from studies on individuals with PTSD [51,52] and, in turn, suggested dysregulation in the stress response system [47]. Despite experiencing increased stress, basic elements of resilience were highlighted in these caregivers, including strong social support from family and friends and spiritual strength.

Data from this study contributes to our understanding of caregivers’ susceptibility and resilient qualities available for the design of public health policy and actions. This study alerts healthcare providers and public health officials to the unique needs of children with developmental and social determinants of health vulnerabilities to promote culturally sensitive actions to support family health and well-being. Strong systemic family support can reduce the caregivers’ burden and mitigate the cumulative trauma that might impact parents’ and children’s health and developmental outcomes.

## Figures and Tables

**Figure 1 ijerph-20-01838-f001:**
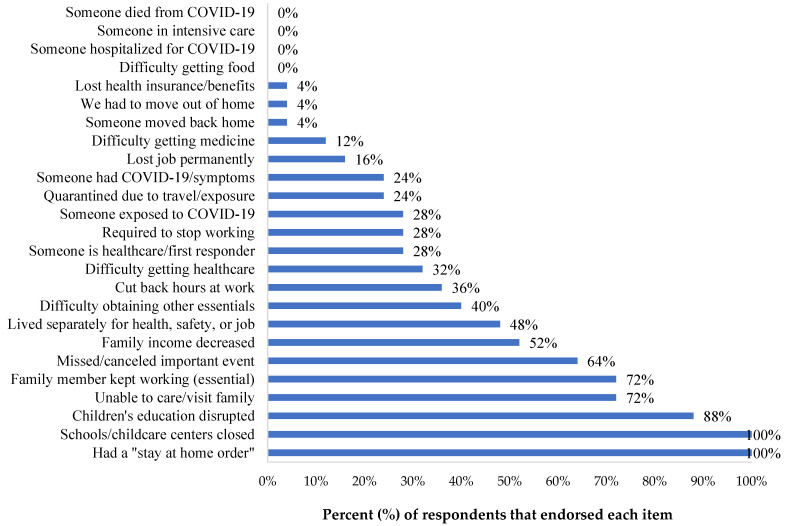
COVID-19 Exposure and Family Impact Survey (CEFIS)- Exposure Scale endorsed items by study participants.

**Figure 2 ijerph-20-01838-f002:**
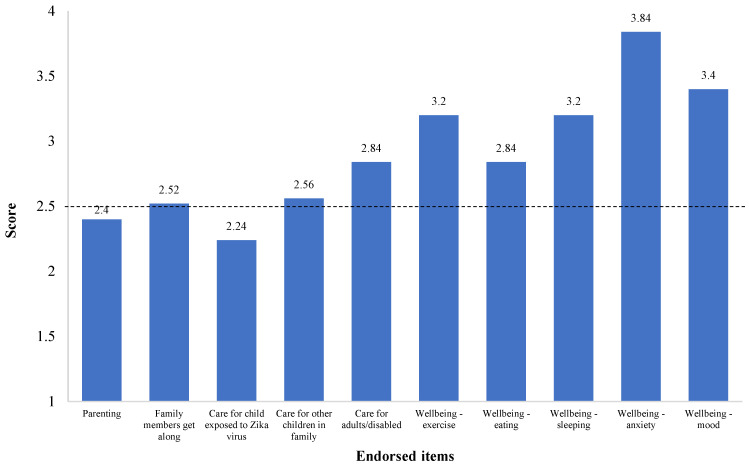
COVID-19 Exposure and Family Impact Survey (CEFIS)—Impact Scale endorsed items (mean values) by study participants. Note: Higher scores indicate a higher impact with worse outcomes. The midpoint of the scale (identified with a dotted line) is 2.5; the items’ score range is 1 to 4 points.

**Figure 3 ijerph-20-01838-f003:**
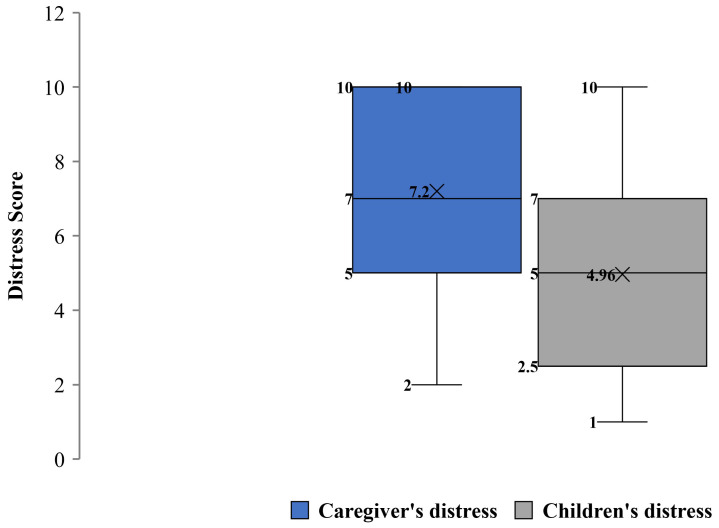
COVID-19 Exposure and Family Impact Survey (CEFIS)—Distress Scale experienced by caregivers and their children due to COVID-19. Note: Higher scores indicate higher perceived distress; the items score range is 1 to 10 points.

**Figure 4 ijerph-20-01838-f004:**
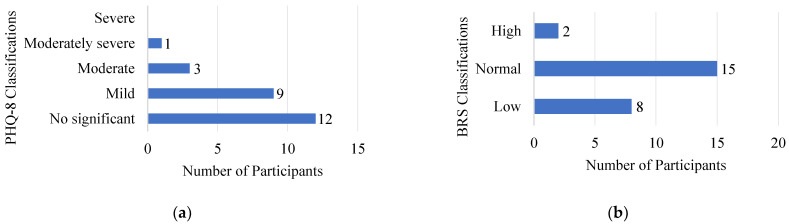
Caregivers’ scores classification on (**a**) the Patient Health Questionnaire (PHQ-8) and (**b**) the Brief Resilience Scale (BRS).

**Figure 5 ijerph-20-01838-f005:**
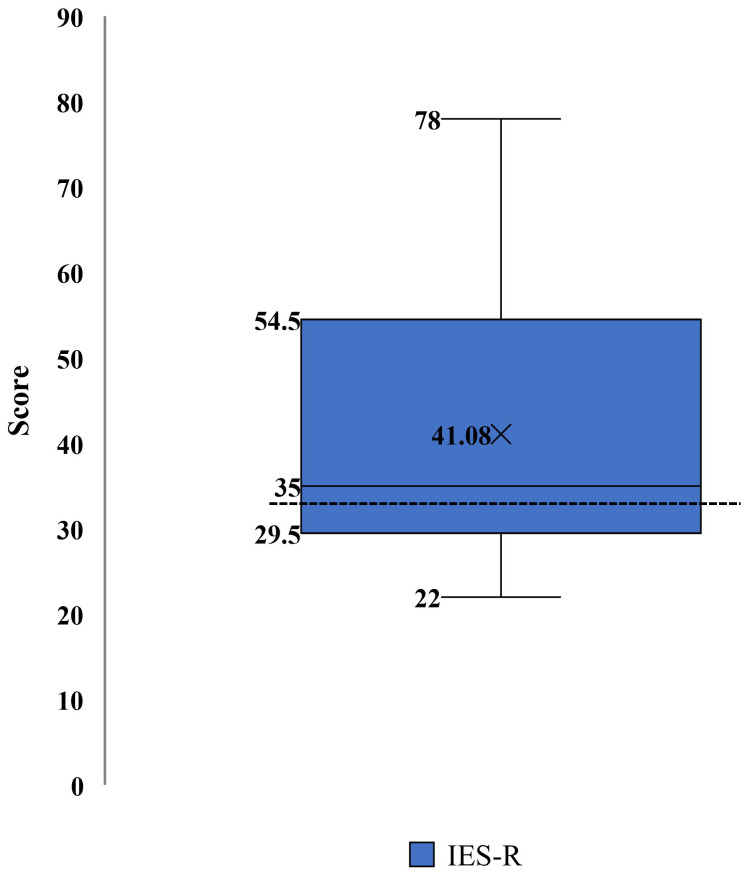
Impact of Event Scale-Revised (IES-R) scores distribution for post-traumatic stress symptomatology in caregivers. Note: Higher scores indicate higher post-traumatic stress symptomatology; items score ranges from 0 to 88 points. A score of 33+ (identified with a dotted line) suggests a probable post-traumatic stress disorder (PTSD) diagnosis.

**Table 1 ijerph-20-01838-t001:** Characteristics of Study Participants.

Caregiver’s Characteristics	n (%)
Age in years, mean ± standard deviation	30.4 ± 6.59
Sex, female	23 (92.0)
Household income	
Less than $10,000	12 (48.0)
$10,000–$14,999	7 (28.0)
$15,000–$24,999	4 (16.0)
$35,000–$49,999	2 (8.0)
Educational attainment	
High school	8 (32.0)
Technical/occupational degree	11 (44.0)
Associate or bachelor’s degree	6 (24.0)
Employment status, not working	17 (68.0)
Marital status	
Married/cohabitating	15 (60.0)
Single	8 (32.0)
Other *	2 (8.0)
Health insurance	
Public	20 (80.0)
Private	4 (16.0)
None	1 (4.0)
Race	
White	5 (20.0)
Other [Mexican, Puerto Rican, Cuban, Dominican, South American]	19 (76.0)
“Don’t know”	1 (4.0)
History of mental health problems	
Anxiety or depression	3 (12.0)
None	22 (88.0)
Food security indicator	
High or marginal food security	15 (60.0)
Low or very low food security	10 (40.0)
Participants’ POPZE child characteristics	
Age in years, mean ± standard deviation	2.8 ± 0.71
Sex, female	15 (60.0)
Race	
White	5 (20.0)
Other [Mexican, Puerto Rican, Cuban, Dominican, South American]	19 (76.0)
“Don’t know”	1 (4.0)
Had developmental delay ^#^ at 24 months (n = 21)	13 (61.9)
Receive specialized services ^†^	13 (52.0)
Interruptions on specialized services	6 (46.2)

* Divorced or separated. ^#^ Cognitive, language, or motor delay, measured by professional assessment with the Bayley Scales of Infant and Toddler Development, Third Edition (BSID-III). **^†^** Speech, occupational, physical, or psychological therapy.

**Table 2 ijerph-20-01838-t002:** Correlations among study variables.

	1	2	3	4	5	6
1. HCC (pg/mg)	1					
2. CEFIS Exposure Scale	0.32	1				
3. CEFIS Impact Scale	0.31	0.23	1			
4. PHQ-8 scores	0.22	0.53 **	0.20	1		
5. IES-R scores	−0.16	0.26	0.23	0.43 *	1	
6. BRS scores	0.04	−0.31	−0.18	−0.56 **	−0.43 *	1

Note: HCC = hair cortisol concentration; CEFIS = COVID-19 Exposure and Family Impact Survey; PHQ-8 = 8-item Patient Health Questionnaire; IES-R = Impact of Event Scale-Revised; BRS = Brief Resilience Scale. * *p* < 0.05. ** *p* < 0.01.

**Table 3 ijerph-20-01838-t003:** Adjusted linear regression models for the association between stressors and hair cortisol concentrations.

Stressor	Beta	SE	*p*-Value
IES-R continuous score	−0.02	0.02	0.29
IES-R classification—possible trauma	−1.11	0.51	0.04
Food insecurity	1.01	0.48	0.049

Note: All models controlled for caregivers’ age, education, and child sex. Models do not control for other stressors listed in the table. HCC was log-transformed due to a skewed distribution. The linear regression model had continuous HCC as the outcome.

## Data Availability

The data presented are available on request from the corresponding author.

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
