# Peer review of "Stressors and Impact of the COVID-19 Pandemic on Vulnerable Hispanic Caregivers and Children"

_ijerph, 2023, doi:10.3390/ijerph20031838_

Round 1

Reviewer 1 Report

- Authors should expand the Introduction in terms of the global disruptions that the pandemic constituted for societies across different areas of life (doi.org/ 10.3390/su14159699). Here authors should provide more specific insights to the population explored from a societal perspective (doi.org/10.1016/j.jnma.2021.12.010).

-Authors may want to address how their methodology controls for caregiver exposure to non-COVID19 stress factors or how PTSD symptomology could be solely attributed to COVID-19 disruptions. 

- Authors may want to address whether their results could be considered above or below expected levels for this specific vulnerable social group. This may provide additional insights on the impact of COVID-19 disruptions.

Reviewer 2 Report

This is a well-written paper describing the impact of COVID-19 on Hispanic caregivers in Puerto Rico and their children who had prenatal exposure to Zika virus. The authors conducted a cross-sectional study using validated survey instruments and supplemented with measurements of hair cortisol. Their results can inform public health decisions regarding the need for increased social support for populations with SDoH hardships and facing multiple types of stressors.

Strengths:

  • The study population is very unique in that residents of Puerto Rico have endured significant stress from several natural disasters in addition to the COVID-19 pandemic, and this study enrolled participants who have the added health burden of Zika virus exposure while pregnant and are now providing care to their children
  • The methods are clearly explained and make use of several validated instruments
  • The addition of hair cortisol levels provides additional insights into the impact of these continued stressors
  • Data are presented clearly
  • The discussion helps set the results of this study in context with the literature and proposes logical explanations for the current results

Weaknesses (minor):

  • Lines 83-96: the data were collected in 2022 but the IRB approval date appears to be August 26, 2021. Maybe this August 2021 date is the expiration date of the approval? It should be clarified that data were collected after, not before, receiving IRB approval.
  • Lines 130-134: The scoring for the PHQ-8 is described but the reference provided (reference # 21) for scoring and cutoff for depression is for PHQ-9 (not PHQ-8). Please correct if this is inaccurate.

Reviewer 3 Report

Thank you for the opportunity to review this manuscript. The authors explored the social determinants of stress and risk for mental health problems for caregivers of children with prenatal exposure to the Zika virus. Participants included 25 Hispanic caregivers who completed surveys that measures SDoH vulnerabilities, COVID-exposures and impact, and PTSD symptomology. In addition, participants provided hair samples to measure cortisol levels. Results indicated multiple stressors (children’s education, specialized services, food insecurity), highly varied perceptions of COVID-19 impact on daily life, and high number of reports of PTSD symptoms.

Overall, I appreciate the authors’ work and believe it may be of interest to readers. However, there are numerous issues/questions that must be addressed before consideration of publication:

Given the focus on a survey that assesses for symptoms of PTSD, a discussion about this and how you came to focus on symptoms of PTSD in the introduction section of the paper is warranted.

Elaborate further on how supporting caregivers is a way to “protect children.” There is research that links the wellbeing of caregivers and children. Please provide some of this background information to fully set the context for readers.

Regarding the examination of HCC, how accurate/reliable is this method for assessing cortisol levels? The inclusion of this variable requires further justification on theoretical and/or practical grounds.

There are multiple language/grammatical errors throughout. For example, the text in lines 66-68 does not form a complete sentence. These issues significantly detract from the overall quality of this paper. Thorough copyediting is needed.

I wish the authors all the best in their continued work in this meaningful area of research.

Round 2

Reviewer 1 Report

The authors have satisfactorily addressed my initial concerns.